# Synergetic Effects of Intronic Mature miR-944 and ΔNp63 Isoforms on Tumorigenesis in a Cervical Cancer Cell Line

**DOI:** 10.3390/ijms21165612

**Published:** 2020-08-05

**Authors:** Jungho Kim, Sunyoung Park, Yunhee Chang, Kwang Hwa Park, Hyeyoung Lee

**Affiliations:** 1Department of Biomedical Laboratory Science, College of Health Sciences, Catholic University of Pusan, Busan 46252, Korea; jutosa70@cup.ac.kr; 2Department of Biomedical Laboratory Science, College of Health Sciences, Yonsei University, Wonju 26493, Korea; angelsy88@gmail.com (S.P.); jyhskg@hotmail.com (Y.C.); 3School of Mechanical Engineering, Yonsei University, Seoul 03772, Korea; 4Department of Pathology, Yonsei University, Wonju College of Medicine, Wonju 26426, Korea; abba@yonsei.ac.kr

**Keywords:** cervical cancer, miR-944, ΔNp63, EMT

## Abstract

*miR-944* is located in an intron of the tumor protein p63 gene (*TP*63). *miR-944* expression levels in cervical cancer tissues are significantly higher than in normal tissues and are associated with tumor size, International Federation of Gynecology and Obstetrics (FIGO) stage, lymph node metastasis, and survival. However, associations of miR-944 with its host gene, *TP*63, which encodes TAp63 and ΔNp63, in cervical cancer have not been fully investigated. A positive correlation between *miR-944* and *ΔNp63* mRNA expression was identified in cervical cancer tissues. Furthermore, when the expression of *miR-944* and *ΔNp63* was simultaneously inhibited, cell proliferation-, differentiation- epithelial-mesenchymal transition (EMT)-, transcription-, and virus-associated gene clusters were shown to be significantly more active according to functional annotation analysis. Cell viability and migration were more reduced upon simultaneous inhibition with anti-miR-944 or ΔNp63 siRNA than with inhibition with anti-miR-944 or ΔNp63 siRNA alone, or scramble. In addition, Western blot analysis showed that the simultaneous inhibition of *miR-944* and *ΔNp63* reduced EMT by increasing the expression of epithelial markers such as claudin and by decreasing mesenchymal markers such as N-cadherin and vimentin. Slug, an EMT transcription factor, was also decreased by the simultaneous inhibition of *miR-944* and *ΔNp63*. Thus, associations between *miR-944* and *ΔNp63* in cervical cancer could help to elucidate the function of this intronic microRNA and its role in carcinogenesis.

## 1. Introduction

MicroRNAs (miRNAs and miR-), key members of the family of noncoding RNAs, are approximately 18–25 nucleotides in length and are involved in the post-transcriptional regulation of gene expression by either oncogenic (onco-miR) or tumor suppressive functions (tumor suppressive-miRs) [1]. Numerous studies on cancer have demonstrated that miRNAs regulate various carcinogenesis processes such as cell maturation, proliferation, migration, invasion, autophagy, apoptosis, and metastasis [2,3,4]. Therefore, miRNAs are promising potential markers for diagnosis and prognosis and for personalized targeted therapies in cervical cancer patients.

miRNAs are mainly categorized into two classes based on the location of their generation: (1) intergenic miRNAs, generated from the transcripts of miRNA genes located between protein-coding genes, and (2) intragenic miRNAs, generated from the transcripts of sequences such as introns and exons located within protein-coding genes [5,6]. Intergenic miRNAs are known to be transcribed as independent transcription units. Most intragenic miRNAs are located in the introns of host genes and are called intronic miRNAs [7,8,9]. These intronic miRNAs have two means of originating from host gene introns. Some intronic miRNAs are transcribed in the same direction as that of their host genes. However, other intronic miRNAs are independently transcribed, with their own promoters or transcription start sites [10,11]. Some intronic miRNAs are transcribed in the same direction and subsequently are co-expressed with their host genes, and most intronic miRNAs act similarly to the function of their host genes by the degradation of transcripts inhibiting the host genes in host gene-related pathways [12,13]. Other intronic miRNAs are co-expressed with their host genes but their function is opposite to that of their host genes, resulting in negative feedback by binding to the 3′-UTRs of their host genes [10,14].

When intronic miRNAs are expressed independently from host genes by transcription initiation regions, patterns of discordant expression between host genes and intronic miRNAs appear. To better understand the relationships between intronic miRNAs and their host genes, more experimental and clinical validation studies are needed. 

*TP*63, the host gene of *miR-944*, encodes multiple domains, including transactivation (TA), DNA-binding, and oligomerization domains, with a C-terminus and features that have similar modular structures and homology to p53. The *TP*63 gene gives rise to TAp63 containing the TA domain and is generated by the use of an alternative promoter, P1. Similarly to p53, the main functional activity of TAp63 is as a tumor suppressor, which functions by inducing apoptosis, cell cycle arrest, and cellular senescence [15,16]. However, an alternative promoter (P2)-generated N-terminal truncated isoform (ΔNp63) lacking the TA domain regulates either transcriptional activator activity or repressor function by directly binding DNA [15,16]. ΔNp63 functions as either a tumor suppressor or as an oncogene, maintaining development, stem cell renewal, and cellular differentiation during normal homeostasis. In cancer, the over-expression of ΔNp63 is reported to be related to cell survival, as well as cell migration and the epithelial-to-mesenchymal transition (EMT)-related features of cancer cells [17,18]. However, several studies have reported that the down-regulation of ΔNp63 is also related to moderately or poorly differentiated tumors and to poor prognosis [19,20].

*miR-944* is reported to be independently expressed by its own promoter in normal human keratinocytes. The ΔNp63 protein directly binds to the *miR-944* promoter and drives transcription in normal keratinocytes [21]. However, usually, intronic miRNAs are transcribed in the same direction as the host gene and are expressed together. It is not fully understood whether *miR-944* expression is associated with either *TAp63* or *ΔNp63*, and the role of intronic *miR-944* in cervical cancer circumstances remains unclear. Therefore, the correlations between intronic miR-944 and its host gene, *TP*63 (*TAp63* or *ΔNp63*), in cervical cancer should be investigated to better elucidate the generation of intronic miRNAs and their function in such malignancies. Here, apoptosis, cell migration, and EMT phenomena were investigated in the cervical cancer cell line ME-180 by designing inhibitors of *miR-944* and *ΔNp63*, either alone or together.

## 2. Results

### 2.1. Association between Expression Levels of miR-944 and TAp63 mRNA and ΔNp63 mRNA in Cervical Cancer Tissue

To investigate whether *miR-944* interacts with *p63*, which is the miR-944 host gene, the expression levels of two major isoforms of *TAp63* and *ΔNp63* mRNAs were compared with miR-944 expression levels. The mRNA expression levels of *TAp63* and *ΔNp63* were divided into negative and positive groups according to a cut-off value identified by receiver operator characteristic (ROC) curve analysis in normal and cervical cancer tissues [22]. Hence, the expression levels of *miR-944* were compared in the *TAp63*- and *ΔNp63*-positive and negative groups. The expression levels of *miR-944* were not shown to be significantly related to *TAp63* expression status (mean value: 73.84 for *TAp63* (−), 109.30 for *TAp63* (+), *p* = 0.67) (Figure 1A), while it was significantly over-expressed in the *ΔNp63*-positive group (mean value: 9.78 for *ΔNp63* (−), 98.95 for *ΔNp63* (+), *p* = 0.01) (Figure 1B). Moreover, although the expression levels of *miR-944* and *TAp63* were not correlated (Spearman correlation coefficient *r* = −0.1 (95% CI (−0.3)–0.2, *p* = 0.56)) (Figure 1C), the expression levels of *miR-944* and *ΔNp63* showed a significant positive correlation (*r* = 0.34 (95% CI 0.06–0.51, *p* = 0.001)) (Figure 1D).

### 2.2. Cell Viability According to Inhibition of miR-944 and ΔNp63 in ME-180 Cells

To screen for inhibitory effects of *miR-944* and *ΔNp63* and to explore their role in cell proliferation, anti-miR-944 and ΔNp63 siRNA at concentrations of 25, 50, and 100 nM were transfected into the ME-180 cell line for 24 h. The expression levels of *miR-944* and *ΔNp63* were confirmed by reverse transcription quantitative PCR (RT-qPCR). The expression of *miR-944* and *ΔNp63* in ME-180 transfected with anti-miR-944 and ΔNp63 siRNA was significantly inhibited compared to that in ME-180 transfected with negative control (NC) (*p* < 0.01 and *p* < 0.01, respectively) (Appendix A).

Subsequently, cell viability was measured by the water-soluble tetrazolium salt (WST) assay. Anti-miR-944 induced a decrease in cell viability in both a dose-dependent manner and time-dependent manner (Figure 2A,C). In addition, when *ΔNp63* was knocked out, cell viability was decreased (Figure 2B,D).

Considering that the cell viability with both anti-miR-944 and ΔNp63 siRNA treatments at 100 nM was significantly diminished, a subsequent loss-of-function study of *ΔNp63* and *miR-944* in ME-180 cells was performed at a final total concentration of 100 nM, comprising 50 nM of anti-miR-944 and ΔNp63 siRNA. 

### 2.3. Combined miR-944 and ΔNp63 Inhibition Increases ME-180 Cell Apoptosis

To explore the possible effects of a combination of anti-miR-944 and ΔNp63 siRNA, ME-180 cells transfected with anti-miR-944, ΔNp63 siRNA, and a combination of anti-miR-944 and ΔNp63 siRNA at a final concentration of 100 nM for 24, 48, and 72 h were investigated for cell viability (Figure 3A). For the WST assays at 24 h, all anti-miR-944 (83%), ΔNp63 siRNA (80%), and combined anti-miR-944 and ΔNp63 siRNA (74%) treatments significantly decreased cell viability compared to that in those transfected with scramble (100%) (*p* < 0.001, *p* < 0.001, and *p* < 0.001, respectively). The combined anti-miR-944 and ΔNp63 siRNA transfection was found to result in significantly lower cell viability compared to that of cells transfected with anti-miR-944 and ΔNp63 siRNA separately at 24, 48, and 72 h (Figure 3B). In addition, the result of the trypan blue assay also showed that the combination of anti-miR-944 and ΔNp63 siRNA increased the growth inhibitory effects on ME-180 cells compared to those from treatment with anti-miR-944 or ΔNp63 siRNA alone (Figure 3C).

Since decreased cell viability was found for anti-miR-944, ΔNp63 siRNA, and the combination of anti-miR-944 and ΔNp63 siRNA, we further sought to identify whether the inhibition with anti-miR-944, ΔNp63 siRNA, and the combination of anti-miR-944 and ΔNp63 siRNA induced cell apoptosis. To this end, ME-180 cells transfected with anti-miR-944, ΔNp63 siRNA, and a combination of anti-miR-944 and ΔNp63 siRNA were stained with propidium iodide (PI) and Annexin V, and the percentage of apoptotic cells was analyzed by flow cytometry (Figure 4A). The percentage of apoptotic ME-180 cells transfected with anti-miR-944 (17.8%), ΔNp63 siRNA (22.0%), and a combination of anti-miR-944 and ΔNp63 siRNA (29.3%) was found to be significantly increased compared with that of cells transfected with scramble (8.6%) (*p* < 0.01, *p* < 0.001, and *p* < 0.001, respectively). Cells transfected with a combination of anti-miR-944 and ΔNp63 siRNA showed increased apoptosis compared with cells transfected with anti-miR-944 and ΔNp63 siRNA individually (*p* < 0.05 and *p* < 0.001, respectively) (Figure 4B).

To confirm apoptosis, the levels of cleaved poly ADP-ribose polymerase (PARP), which functions in inducing apoptosis, were measured. Cleaved PARP protein was found to be increased in ME-180 cells transfected with anti-miR-944, ΔNp63 siRNA, and a combination of anti-miR-944 and ΔNp63 siRNA, as compared to in those transfected with scramble (Figure 4C).

### 2.4. Combined miR-944 and ΔNp63 Inhibition Decreases Migration in ME-180 Cells

To evaluate possible anti-cancer activities, wound healing and transwell invasion assays were performed. After ME-180 cells were transfected with anti-miR-944, ΔNp63 siRNA, and a combination of anti-miR-944 and ΔNp63 siRNAs, cells were scratched in 500 μm widths and wound closure by cellular migration was measured at 12 and 24 h using the ImageJ software (Figure 5A). The ability to close wounds of the ME-180 cells transfected with anti-miR-944, ΔNp63 siRNA, and a combination of anti-miR-944 and ΔNp63 siRNA at 24 h was significantly decreased by 80.6%. 42.2%, and 30.2%, respectively, as compared to that of cells transfected with scramble at 24 h (*p* < 0.05, *p* < 0.001, and *p* < 0.001, respectively) (Figure 5B). Moreover, in the transwell assays, the percentage of invading cells was significantly decreased by anti-miR-944 (60.7%) and ΔNp63 siRNA (40.7%) alone (*p* < 0.05 and *p* < 0.001, respectively), and the lowest percentage of invading cells was observed with combined *miR-944* and *ΔNp63* inhibition (35.3%) (*p* < 0.001), as compared to that with scramble (100%) (Figure 5C,D).

### 2.5. Combined miR-944 and ΔNp63 Inhibition Affects EMT in ME-180 Cells

Subsequently, we explored whether *miR-944*, *ΔNp63*, and combined *miR-944* and *ΔNp63* inhibition play a role in regulating the process of EMT, which is closely related to malignant characteristics. Western blot assays were conducted to compare the protein levels of EMT markers. For the epithelial markers E-cadherin and claudin-1, while E-cadherin was not found to have any significant difference, claudin-1 increased in ME-180 cells transfected with anti-miR-944, ΔNp63 siRNA, and the combined anti-miR-944 and ΔNp63 siRNA, compared to in those transfected with scramble, respectively (Figure 5E). By contrast, the mesenchymal marker N-cadherin was decreased in ME-180 cells transfected with ΔNp63 siRNA and combined anti-miR-944 and ΔNp63 siRNA, as compared to in those transfected with scramble, respectively. Furthermore, ME-180 cells transfected with a combination of anti-miR-944 and ΔNp63 siRNA were found to exhibit decreased N-cadherin abundance, compared to those transfected with anti-miR-944 and ΔNp63 siRNA individually. Vimentin was decreased in ME-180 cells transfected with the combined anti-miR-944 and ΔNp63 siRNA, as compared to scramble (Figure 5E).

To further investigate whether transcription factors inducing EMT were affected by combined *miR-944* and *ΔNp63* inhibition, the levels of slug were examined. According to the results of Western blot analysis, slug was also decreased in cells transfected with combined anti-miR-944 and ΔNp63 siRNA. Taken together, the combination of *miR-944* and *ΔNp63* inhibition could enhance the repression of EMT in cervical cancer.

### 2.6. Identification of Differentially Expressed Genes (DEGs) According to miR-944, ΔNp63, and Combined miR-944 and ΔNp63 Inhibition

To gain insight into the underlying mechanisms and investigate the association of combined *miR-944* and *ΔNp63* inhibition with gene expression profiles, RNA sequence analysis was performed using total RNA extracted from ME-180 cells transfected with scramble, anti-miR-944, ΔNp63 siRNA, and combined anti-miR-944 and ΔNp63 siRNA. The selection criteria for DEGs were as follows: |log2 fold change| ≥ 1.5 (*p* < 0.05, false discovery rate (FDR) < 0.05). A heat-map profile of mRNA expression displaying differentially regulated transcripts is shown in Figure 6A. Compared to scramble, the numbers of significantly DEGs were 206 genes in ME-180 cells treated with anti-miR-944 (blue), 434 genes in ME-180 cells treated with ΔNp63 siRNA (yellow), and 1338 genes in ME-180 cells treated with combined anti-miR-944 and ΔNp63 siRNA (red) (Figure 6B). The Venn diagrams for common DEGs in ME-180 cells treated with anti-miR-944, ΔNp63 siRNA, and combination of anti-miR-944 and ΔNp63 siRNA are shown in Figure 6C.

Furthermore, to establish functional features, Gene Ontology (GO) terms involving the DEGs were further determined using the Database for Annotation, Visualization and Integrated Discovery (DAVID) tool. The top GO-enriched biological processes for each of the three inhibition conditions (inhibition with anti-miR-944 and ΔNp63 siRNA, separately and combined) were clustered with the ClueGO + CluePedia software (kappa score  =  0.4, *p* < 0.05 with Bonferroni step-down analysis). The top GO-enriched terms are listed in Appendix A. Among the GO terms, the top four clusters were classified into cell-proliferation-associated (Cluster 1), cell-differentiation-and-EMT-associated (Cluster 2), transcription-associated (Cluster 3), and virus-associated (Cluster 4). Significant GO terms in terms of *miR-944* inhibition were identified as cell death, cell–cell adherence junctions, and nucleosome assembly (Figure 6D). Significant GO terms for *ΔNp63* inhibition were listed as regulation of the apoptotic process, epithelial cell differentiation, epidermis development, bicellular tight junctions, cell development, signal transduction, transcription factor activity, sequence-specific DNA binding transcription factor activity, and responses to viruses (Figure 6E). The significant GO terms for the combination of *miR-944* and *ΔNp63* inhibition were listed as negative regulation of cell proliferation, positive regulation of apoptotic processes, negative regulation of growth, epidermis development, regulation of cell motility, wound healing, cell–cell signaling, regulation of cell migration, regulation of transcription from RNA polymerase II promoters, DNA replication-dependent nucleosome assembly, positive and negative regulation of gene expression, epigenetic, gene silencing, type I interferon signaling pathway, defense response to viruses, innate immune response in mucosa, and response to viruses (Figure 6F).

## 3. Discussion

In the present study, the correlation between *miR-944* and its host gene, *TP*63, which encodes TAp63 and ΔNp63, and its role in carcinogenesis were examined. The results of correlation analysis showed that in cervical cancer tissues, intronic *miR-944* was co-expressed with *ΔNp63* but not with *TAp63* (Figure 1). Intronic miRNAs are often reported to be functionally related to their host genes by the degradation of transcripts in the host-gene-related pathway. Therefore, it was hypothesized that *miR-944* may be functionally connected to its host gene, *TP*63, especially *ΔNp63*, in cervical cancer. Previous studies showed that *ΔNp63* mRNA expression levels were significantly higher in the ME-180, SiHa, CaSki, and C33A cell lines than *TAp63* mRNA expression levels and that *miR-944* expression levels in ME-180, CaSki, and C4I were higher than those in other cervical cancer cell lines such as HeLa, SW756, and C33A [22,23].

Firstly, apoptosis and migration induced by the inhibition of *miR-944* and *ΔNp63* in cervical cancer was investigated. The inhibition of *miR-944* in ME-180 cervical cancer cells induced apoptosis and suppressed migration (Figure 2, Figure 3 and Figure 4). The inhibition of *miR-944* in Ishikawa endometrial cells prominently prevents cell cycle progression and induces cancer cell apoptosis [24]. *miR-944* inhibition suppresses cell migration and invasion in the breast cancer cell lines MDA-MB 231 and MCF-7 [25]. In addition, gain- and loss-of-function experiments with *miR-944* revealed the effects of migration and invasion in Caski and HeLa cervical cancer cells [23]. Furthermore, the expression levels of *miR-944* in cervical tissues were significantly higher than those in normal tissues, and high *miR-944* expression was associated with bulky tumor size, advanced stage, lymph node metastasis, and poor survival [23,26]. These results support the idea that *miR-944* acts as an onco-miR by regulating apoptosis and promoting cell migration and invasion.

The inhibition of *ΔNp63* in ME-180 cervical cancer cells induced apoptosis and increased the level of cleaved PARP protein. In addition, the inhibition of *ΔNp63* in such cells suppressed migration. The results of RNA sequence analysis in the present study also supported the notion that the inhibition of *ΔNp63* is involved in the regulation of apoptotic processes. Several previous studies have reported that p63 reduction leads to apoptosis, as evidenced by increased cleaved PARP levels, augmented Annexin V staining, and elevated PUMA and NOXA expression [27,28,29]. ΔNp63 in breast and esophageal squamous carcinoma was previously reported to act as an activator of cell migration and invasion [30,31]. ΔNp63 was also highly expressed in several cancer tissues such as esophageal squamous cell carcinoma and cervical cancer with poor prognosis [22,32]. Similarly, our results also demonstrated that *ΔNp63* induces apoptosis and activates migration and invasion in cervical cancer cells.

Based on the results of flow cytometric analysis, the present study showed that dual inhibition by anti-miR-944 and ΔNp63 siRNA induced more apoptosis than *miR-944* and *ΔNp63* inhibition alone (Figure 3). Functional annotation through RNA sequence analysis demonstrated that DEGs with the combination of *miR-944* and *ΔNp63* inhibition were involved in the negative regulation of cell proliferation, positive regulation of apoptotic processes, and negative regulation of growth (Figure 6). Moreover, based on the wound healing and transwell assays, the combination of *miR-944* and *ΔNp63* inhibition decreased migration and invasion to a greater extent than *miR-944* and *ΔNp63* inhibition alone (Figure 5). Moreover, the GO terms of DEGs for the combination of *miR-944* and *ΔNp63* inhibition were involved in epidermis development, the regulation of cell motility, wound healing, cell–cell signaling, and the regulation of cell migration. Taken together, these results demonstrated that dual inhibition by anti-miR-944 and ΔNp63 siRNA may effectively regulate cellular apoptosis, migration, and invasion in cervical cancer.

Cell migration and invasion were related to EMT’s molecular mechanisms. Claudin, a representative epithelial marker, was shown to be significantly increased not only with *miR-944* and *ΔNp63* inhibition alone but also with the combination of *miR-944* and *ΔNp63* inhibition, as compared to with scramble (Figure 5). Claudins, which are epithelial markers, are transmembrane proteins and important components of tight junctions (TJs), which are central to the regulation of paracellular permeability and for the maintenance of epithelial cell polarity [33]. The results reported by Kaneko et al., are consistent with our findings in that the knock-down of *ΔNp63* in human nasal epithelial cells enhances barrier and fence functions, inducing claudin-1 and -4 and inhibiting p38 MAPK [34]. However, no change was observed in E-cadherin, which is another epithelial marker. A typical type of EMT in malignant progression involves cadherin switching, which indicates decreased levels of E-cadherin and increased N-cadherin abundance [35]; however, it is reported that EMT does not always accompany decreased E-cadherin in breast cancer [36]. Interestingly, N-cadherin and vimentin, representative mesenchymal markers, were found to be significantly decreased with the combination of *miR-944* and *ΔNp63* inhibition, compared to with inhibition by anti-miR-944 or ΔNp63 siRNA alone. Several previous reports described that N-cadherin expression is more important for cancer metastasis than E-cadherin and other EMT inducers [37,38]. In addition, the levels of slug protein, with the combination of *miR-944* and *ΔNp63* inhibition, decreased more than with *miR-944* or *ΔNp63* inhibition alone. The transcription factor slug was previously reported to be implicated in the control of EMT. Slug has anti-apoptotic activity, and its levels are increased in patients with breast and ovarian cancers [39]. Furthermore, higher levels of slug expression have been reported in more aggressive forms of breast cancer such as basal-like breast carcinoma [40]. EMT leads to chemo-resistance and radio-resistance in cervical cancer cells. Inhibiting EMT in cervical cancer cells sensitizes them to radiation and drugs, leading to the increased survival of cervical cancer patients [35]. These results provide meaningful information for further research regarding chemo- and radio-resistance targets in cervical cancer.

In conclusion, the association between *miR-944* and *ΔNp63* in cervical cancer will help to better elucidate the function of this intronic microRNA and its role in carcinogenesis. Additionally, further investigation with various cancer cell lines including that of the correlation between *miR-944* and *ΔNp63* with colony forming assays and in vivo tests is necessary.

## 4. Materials and Methods

### 4.1. Clinical Samples

A total of 66 formalin-fixed paraffin-embedded (FFPE) cervical cancer tissue samples were collected after pathological diagnosis at the Department of Pathology, Yonsei University Wonju Severance Christian Hospital, between January 2010 and December 2014. This study was approved by the institutional ethics committee at Yonsei University Wonju College of Medicine (approval no. CR315052, date: 10 March 2016), and the study participants provided written informed consent.

### 4.2. Cell Culture

The cervical cancer cell line ME-180 was purchased from the Korean Cell Line Bank (Seoul, Korea). The cells were incubated in Roswell Park Memorial Institute (RPMI) medium (Gibco, Carlsbad, CA, USA) with 10% fetal bovine serum (FBS; Gibco) and 1% penicillin/streptomycin (Gibco) at 37°C in a humidified 5% CO_2_ atmosphere.

### 4.3. Transfection of Anti-miRNA and siRNA

Anti-miR-944 and ΔNp63 siRNA, and scramble as a negative control (Bioneer, Daejeon, Republic of Korea), with Lipofectamine 2000 (Thermo Fisher Scientific, Inc., Waltham, MA, USA) were transfected into ME-180 cervical cancer cells. The sequence of the ΔNp63 siRNA is listed in Appendix A. ME-180 cells were seeded into 6-well plates (2 × 10^5^/well), 24-well plates (5 × 10^4^/well), or 96-well plates (5 × 10^3^/well) then transfected with anti-miR-944, ΔNp63 siRNA, and a combination of anti-miR-944 and ΔNp63 siRNA in serum-free medium for 4 h, and then further incubated in medium with 10% FBS for 24 h. Cells were then subjected to further assays or RNA/protein analysis.

### 4.4. Cell Viability Assay Using EZ-Cytox

Cells were subjected to water soluble tetrazolium salt (WST) viability assays using an EZ-Cytox cell viability assay kit (DaeilLab, Seoul, Korea), according to the manufacturer’s recommendations. Ten microliters of WST solution was added to the cell culture media and incubated for 30 min in a CO_2_ incubator. The optical absorbance was measured at 450 nm using an Infinite 200 spectrophotometer (Tecan, Salzburg, Austria).

### 4.5. Trypan Blue Exclusion Assay

ME-180 cells (2 × 10^5^/well) were plated in 6-well plates. After 24 h, cells were transfected with anti-miR-944, ΔNp63 siRNA, and a combination of anti-miR-944 and ΔNp63 siRNA. The cells were incubated for the indicated time periods and subjected to the trypan blue exclusion assay as described previously [41]. Briefly, the cells were trypsinized and trypan blue stain solution (10 μL of 0.4%) was mixed with 10 μL of the trypsinized cells. Non-stained cells were counted using a hematocytometer.

### 4.6. Cell Apoptosis Analysis

Annexin V and propidium iodide (PI) staining was performed using an Annexin V-FITC apoptosis detection kit 1 (BD Biosciences, Sparks, MD, USA) according to the manufacturer’s instructions. Cultured cells were trypsinized, washed twice with cold PBS, then centrifuged at 600 × *g* for 5 min. Thereafter, the cells were resuspended in 500 μL of binding buffer at a concentration of 5 × 10^5^ cells/mL, and 5 μL of Annexin V-FITC and 5 μL of PI were added to the cell suspension. The mixture was incubated for 10 min at 37 °C in the dark and analyzed using a BD FACS Calibur flow cytometer (BD Biosciences) and FlowJo (TreeStar Inc., Ashland, OR, USA).

### 4.7. Wound Healing and Transwell Invasion Assay

Cells were transfected as described above, and cell migration assays were performed with SPLScar^TM^ (SPL, Pocheon, Korea), according to the manufacturer’s instructions. Confluent monolayer cells were linearly scratched using a 500 μm SPLScar^TM^, and the cells were allowed to close the wound. Wound healing images were captured using a light microscope (CKX41, Olympus, Tokyo, Japan) at 0, 12, and 24 h. The percentage of migrated cell areas compared to the initially wounded regions was identified and analyzed using the ImageJ program (National Institutes of Health, Bethesda, MD, USA).

Cell invasion assays were performed with SPLInsert^TM^ Hanging (8 μm, SPL), according to the manufacturer’s instructions. The chambers were inserted into 24-well plates, and 5 × 10^4^ transfected cells were seeded into the upper chambers. Medium containing 0.1% FBS was added to the upper chambers, while medium containing 20% FBS filled the lower wells. Then, the cells were incubated for 48 h at 37 °C in 5% CO_2_. After incubation, the membrane was trimmed and stained with 0.1% crystal violet then observed using a light microscope. Three fields were randomly selected from each membrane, and the number of invading cells was counted.

### 4.8. Protein Isolation and Western Blot Analysis

Transfected cells were washed twice with PBS and lysed in 50 μL of cell lysis buffer containing 1% Triton X-100 (Sigma, St. Louis, MO, USA), protease inhibitor cocktail (Roche, Mannheim, Germany), and PBS. The lysate was centrifuged at 13,000× *g* at 4 °C for 10 min. The supernatants were subjected to Western blotting after SDS-PAGE. The protein samples (15–20 μg) were resolved on 10 or 12% gels and transferred to nitrocellulose membranes (Bio-Rad, Hercules, USA). The membranes were immersed in 5% skim milk in PBS containing 0.5% Tween 20 for 30 min then probed with primary monoclonal antibodies against cleaved PARP (cat. 9541), E-cadherin (cat. 3195), claudin-1 (cat. 13225), N-cadherin (cat. 13116), vimentin (cat. 5741), slug (cat. 9585) (Cell Signaling), and β-actin (cat. Sc-47778) (Santa Cruz, Dallas, TX, USA) overnight at 4 °C (Appendix A). The blots were washed in PBS containing 0.5% Tween 20 and developed with horseradish peroxidase-conjugated secondary anti-mouse (cat. 7076) or anti-rabbit antibody (cat. 7074) (Cell Signaling). The protein signals were enhanced by SuperSignal^®^ WestPico (Thermo Fisher Scientific, Waltham, MA, USA) for visualization. The protein levels were expressed relative to β-actin levels. The densities of the Western bands were measured using the Bio1D software (Vilber Lourmat, Marne la Vallée, France). Uncropped blots for Western blot analysis are shown in Appendix A.

### 4.9. Total RNA Extraction

For total RNA extraction from cervical tissues, three to four FFPE sections of 10 μm thickness were used. To remove the paraffin from the FFPE tissues, 160 μL of Deparaffinization solution (Qiagen, Hilden, Germany) was added and vortexed, followed by incubation for 3 min at 56 °C. RNA extraction was performed using the Qiagen RNeasy FFPE kit (Qiagen) according to the manufacturer’s protocol. Total RNA was isolated from ME-180 cells using Isol-RNA Lysis reagent (Intron Biotechnology, Seoul, Korea), according to the manufacturer’s instructions.

The purity and concentration of the total RNA were determined by measuring the ratio of optical absorbance at 260 and 280 nm using an Infinite 200^®^ plate reader spectrophotometer (Tecan, Männedorf, Switzerland). All preparation and handling procedures were conducted under RNase-free conditions. Isolated total RNA was stored at −80 °C.

### 4.10. TP63 mRNA Expression Analysis

*ΔNp63* and *TAp63* complementary DNA (cDNA) was synthesized using an M-MLV reverse transcriptase kit (Invitrogen, Carlsbad, CA, USA) and random hexamers (Invitrogen) according to the manufacturer’s recommendations. Briefly, 10 μL of total RNA was added to a master mix containing 10 mM dNTPs at neutral pH, 0.25 μg of random hexamers, and 5 μL of diethylpyrocarbonate (DEPC)-treated water. Reactions were incubated at 65 °C for 5 min then chilled on ice. A mixture containing 4 μL of 5 × First-Strand Buffer, 2 μL of 0.1 M dithiothreitol, and 1 μL of M-MLV reverse transcriptase (RT) was added, and cDNA was synthesized at 25 °C for 10 min, followed by 37 °C incubation for 50 min and, finally, 70 °C incubation for 15 min.

The PCR primers and probes for *ΔNp63* and *TAp63* mRNA were synthesized (Bioneer, Daejeon, Korea) (Appendix A). RT-qPCR for *ΔNp63* and *TAp63* was performed using a CFX96 instrument (Bio-Rad). A total volume of 20 μL containing 10 μL of 2 × Thunderbird Probe qPCR mix (Toyobo, Osaka, Japan), 5 μL of primers and TaqMan probe mixture, 3 μL of distilled water, and 2 μL of template cDNA was used. Positive and negative controls were included for each procedure. The reaction conditions for real-time PCR were as follows: 95 °C for 3 min, then 40 cycles at 95 °C for 3 s and 55 °C for 30 s. Glyceraldehyde-3-phosphate dehydrogenase (*GAPDH*) was used as an endogenous control and for the confirmation of mRNA degradation.

mRNA expression levels were quantified by determining cycle threshold (C_T_) values, which represent the number of PCR cycles required for the fluorescence to exceed a value significantly higher than the background fluorescence. The amount of *ΔNp63* and *TAp63* mRNA was determined using the comparative C_T_ method (ΔΔC_T_ method), measuring mRNA relative to an internal housekeeping gene (*GAPDH*) using the CFX Manager Software v1.6 (Bio-Rad) [42]. The equation for the determination of mRNA expression levels was ΔΔC_T_ = 2^−(ΔCT [target sample]−ΔCT [normal sample]^.

### 4.11. miR-944 Expression Analysis

*miR-944* cDNA was synthesized using a TaqMan microRNA Reverse Transcriptase kit (Applied Biosystems by Life Technologies, Foster City, CA, USA) according to the manufacturer’s recommendations. Reaction volumes of 20 μL per sample for cDNA synthesis contained reverse transcriptase (RT) mixture (0.15 μL of 100 mM dNTP mix and 100 mM concentrations each of dATP, dGTP, dCTP, and dTTP at neutral pH), 1 μL of 50 U/μL Reverse transcriptae (RT), 1.5 μL of 10 × RT buffer, 0.19 μL of 20 U/μL RNase inhibitor, and 3 μL of miRNA-specific primer, with the volume adjusted up to 15 μL with nuclease-free water. Then, 10 ng of total RNA of each sample was added. The following TaqMan small RNA assay (Applied Biosystems by Life Technologies) primers were applied: *RNU6B* and *hsa-miR-944*. The cDNA synthesis reactions were performed at 16 °C for 30 min, then 42 °C for 30 min and 85 °C for 5 min.

To investigate *miR-944* expression, RT-qPCR was performed using the TaqMan MicroRNA assay (Applied Biosystems, Foster City, CA, USA) according to the manufacturer’s instructions. A total volume of 20 μL containing 10 μL of 2 × Thunderbird Probe qPCR mix (Toyobo), 1 μL of 20 × primer and TaqMan probe mixture, 7.6 μL of distilled water, and 1.4 μL of template cDNA was used for each sample. The samples were run in duplicate for each experiment. Positive and negative controls were included. No-template controls were included in each run, and these consisted of sterile distilled water instead of template DNA. The qPCRs were performed on a CFX96 Real-Time PCR Detection System (Bio-Rad, Hercules, CA, USA). PCR cycling was 95 °C for 10 min then 40 cycles of 95 °C for 15 s and 60 °C for 60 s.

The amount of *miR-944* was determined using the comparative C_T_ method (ΔΔC_T_ method), measuring microRNA relative to an internal control, *RNU6B*, using the CFX Manager Software v1.6 (Bio-Rad, Hercules, CA, USA). The equation for *miR-944* expression was ΔΔC_T_ = 2^−(ΔCT [target sample]−ΔCT [normal sample])^.

### 4.12. RNA Sequence Analysis

The concentration and quality of the total RNA were checked by means of a Qubit 2.0 fluorometer (Thermo Fisher Scientific, Grand Island, NY, USA). Total RNA (10 ng) was used to prepare strand-specific barcoded RNA libraries with the Ion AmpliSeq^TM^ Transcriptome human gene expression kit (Thermo Fisher Scientific), following the manufacturer’s protocol. Barcoded RNA sequencing (RNA-seq) libraries were quantified by qPCR with a first step at 50 °C for 2 min, then 40 cycles at 95 °C for 20 s, 95 °C for 1 s, and 60 °C for 20 s. Barcoded RNA at a final concentration of 100 pM was pooled and sequenced on the Ion Torrent Proton sequencing platform using 540 chips. The Ion AmpliSeq Transcriptome human gene expression kit is designed for the targeted amplification of over 20,000 human genes simultaneously using a single primer pool. A short amplicon (approximately 110 bp) is amplified for each targeted gene. All the experiments were carried out independently in duplicate.

AmpliSeq sequencing data were obtained using the Torrent Mapping Alignment Program (TAMP) optimized for Ion TorrentTM sequencing data for aligning the raw sequencing reads against a custom reference sequence set containing all transcripts targeted by the AmpliSeq kit [43,44]. mRNA transcripts with fold-change ≥ 1.5, *p* values < 0.05, and FDR < 0.05 were accepted as DEGs using the hg19_Ampliseq_Transcriptome_21K_v1 reference of the transcriptome analysis console (TAC) software (Thermo Fisher Scientific) (Appendix A) [45]. 

### 4.13. Statistical Analysis

Statistical analysis was performed using the GraphPad Prism software v. 6.0 (GraphPad, La Jolla, CA, USA) and SPSS Statistics software v. 21.0 (IBM, Armonk, NY, USA). Correlation analysis between *miR-944* and *TP*63 (*TAp63* and *ΔNp63* mRNA) was performed using Spearman’s correlation coefficient. For the analysis of cell viability, apoptosis, migration, invasion, and protein expression, the comparison for each condition was conducted with the Mann–Whitney U test or ANOVA test with Tukey’s post hoc test. *p*-values < 0.05 were considered statistically significant.

## Figures and Tables

**Figure 1 ijms-21-05612-f001:**
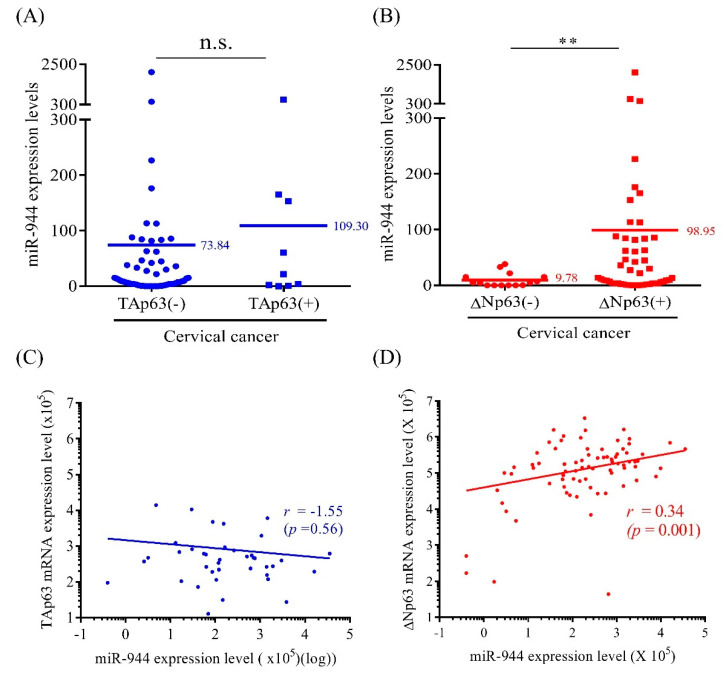
Association between *miR-944* and *p63* isoforms in cervical cancer tissues. *miR-944* expression according to (**A**) *TAp63* expression status and (**B**) *ΔNp63* expression status was analyzed using the Mann–Whitney U test. Correlation of *miR-944* with (**C**) *TAp63* and (**D**) *ΔNp63* was evaluated using Spearman’s rank correlation (*n* = 66). ** *p* < 0.01.

**Figure 2 ijms-21-05612-f002:**
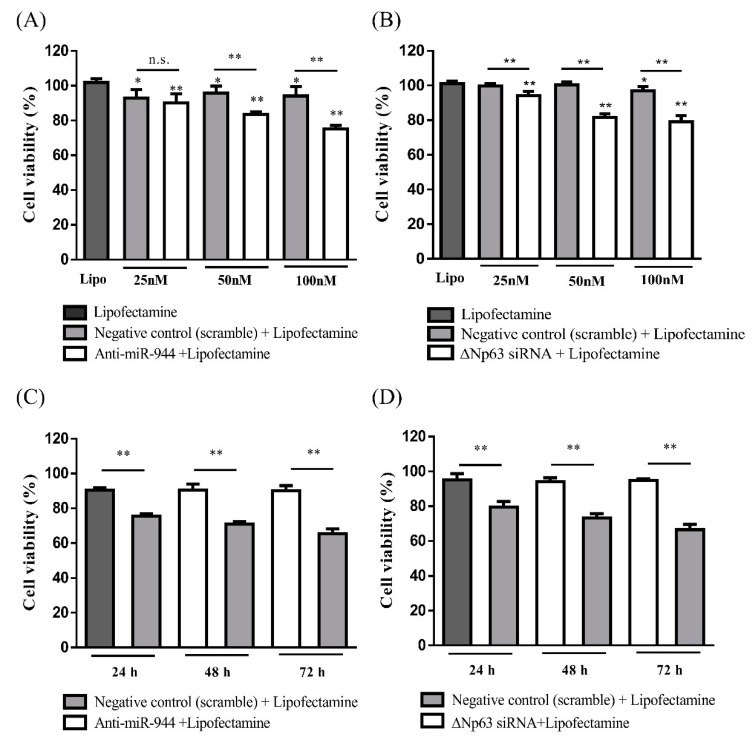
Viability of ME-180 cells after inhibition with anti-miR-944 and ΔNp63 siRNA. ME-180 cells were transfected with (**A**) anti-miR-944 and (**B**) ΔNp63 siRNA for 24 h at 25, 50, and 100 nM. ME-180 cells were treated with 100 nM concentrations of (**C**) anti-miR-944 and (**D**) ΔNp63 siRNA for the indicated periods of time (24, 48, and 72 h). Viable cell counts were measured by the water-soluble tetrazolium salt (WST) assay. Data are reported as means ± SD for five independent experiments and were analyzed using unpaired Mann–Whitney U tests. * *p* < 0.05, ** *p* < 0.01.

**Figure 3 ijms-21-05612-f003:**
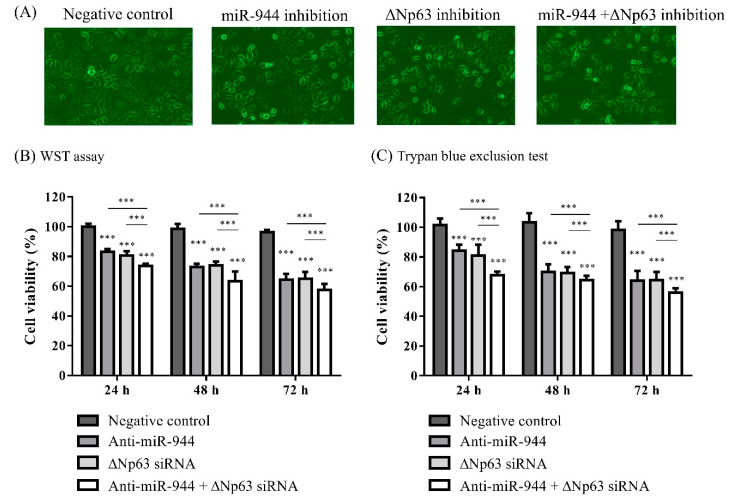
Viability of ME-180 cells after inhibition with anti-miR-944, ΔNp63 siRNA, and a combination of anti-miR-944 and ΔNp63 siRNA. (**A**) ME-180 cells were transfected with anti-miR-944, ΔNp63 siRNA, and a combination of anti-miR-944 and ΔNp63 siRNA at a final concentration of 100 nM for 24, 48, and 72 h and observed with microscopy (200 × magnification). Viable cell counts were measured using (**B**) the WST assay and (**C**) trypan blue exclusion assay. Data are reported as means ± SD for three independent experiments and were analyzed using ANOVA tests with Tukey’s post hoc test. *** *p* < 0.001.

**Figure 4 ijms-21-05612-f004:**
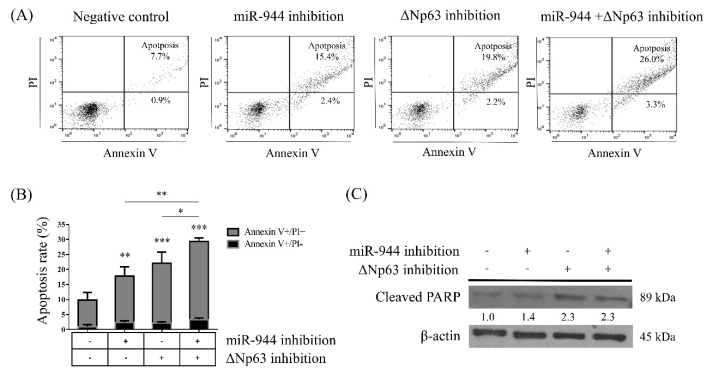
Effects of inhibition with anti-miR-944, ΔNp63 siRNA, and a combination of anti-miR-944 and ΔNp63 siRNA on cell apoptosis. ME-180 cells were transfected with anti-miR-944, ΔNp63 siRNA, and a combination of anti-miR-944 and ΔNp63 siRNA for 24 h. (**A**) After transfection, cells stained with annexin V-Fluorescein isothiocyanate (FITC) and propidium iodide (PI) were analyzed with flow cytometry for determining the apoptotic cell populations. Stained cells were illustrated on the quadrant with the FlowJo software. (**B**) Percentages of apoptotic cells (early and late apoptosis rates) are indicated as a graph. (**C**) Cleaved PARP was analyzed by Western blotting. β-actin was used as an internal control. Data are reported as means ± SD for three independent experiments and were analyzed using ANOVA tests with Tukey’s post hoc test. * *p* < 0.05, ** *p* < 0.01, *** *p* < 0.001.

**Figure 5 ijms-21-05612-f005:**
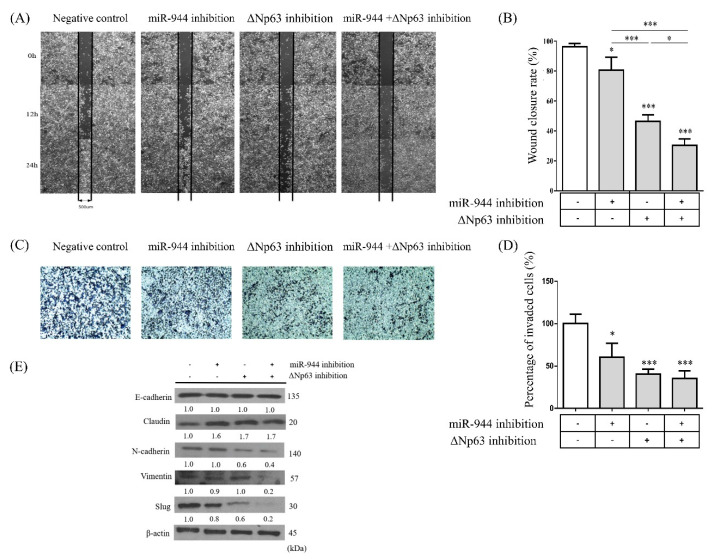
Inhibitory effects of anti-miR-944, ΔNp63 siRNA, and a combination of anti-miR-944 and ΔNp63 siRNA on cell migration. (**A**) After ME-180 cells were transfected with anti-miR-944, ΔNp63 siRNA, and a combination of anti-miR-944 and ΔNp63 siRNA, the cell monolayer was linearly scratched to generate 500 μm wounds, and wound healing closure changes were observed at 12 and 24 h by microscopy. (**B**) The percentage of wound closure was analyzed using the ImageJ software. (**C**) Invasion of ME-180 cells transfected with anti-miR-944, ΔNp63 siRNA, and combination of anti-miR-944 and ΔNp63 siRNA was observed with microscopy. (**D**) The number of invading cells was measured by counting the cells invading through membranes per field. (**E**) The expression of proteins associated with epithelial-to-mesenchymal (EMT) (E-cadherin, claudin, N-cadherin, vimentin, and slug) after *miR-944*, *ΔNp63*, and combined *miR-944* and *ΔNp63* inhibition was analyzed by Western blotting. β-actin was used as a loading control. Data are reported as means ± SD for three independent experiments and were analyzed using ANOVA tests with Tukey’s post hoc test. * *p* < 0.05, *** *p* < 0.001.

**Figure 6 ijms-21-05612-f006:**
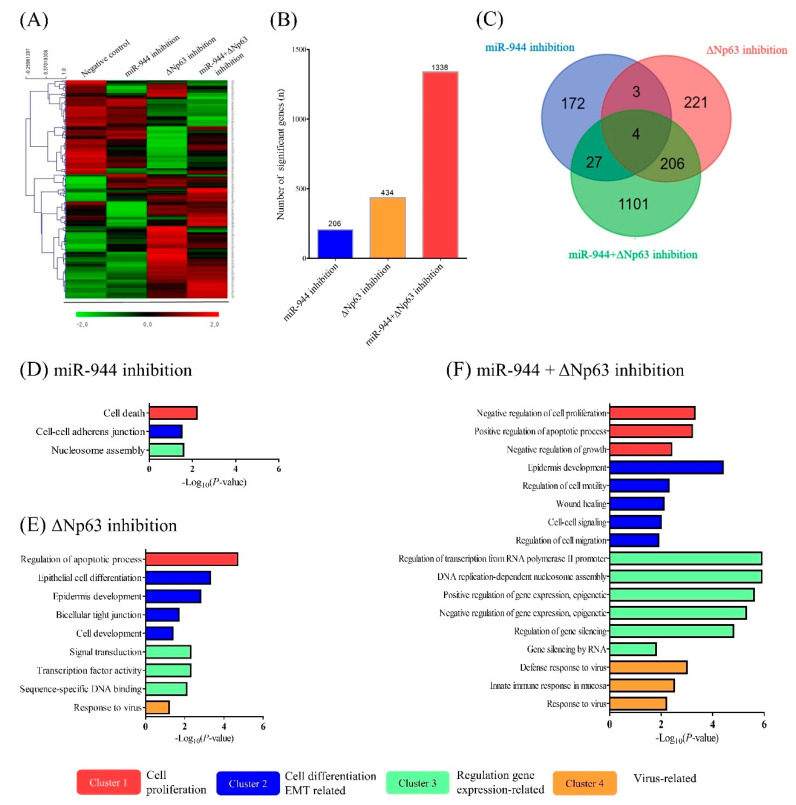
Gene expression profiles after inhibition with anti-miR-944, ΔNp63 siRNA, and a combination of anti-miR-944 and ΔNp63 siRNA. (**A**) Heat map illustrating results for ME-180 cells transfected with anti-miR-944, ΔNp63 siRNA, and a combination of anti-miR-944 and ΔNp63 siRNA. The heat map was obtained by the two-way hierarchical clustering of significantly expressed genes using the MeV software (elucidation distance, *p* < 0.05 by hierarchical clustering analysis). The color scheme is as follows: upregulation in red and downregulation in green. (**B**) Differentially expressed genes (DEGs) in ME-180 cells treated with anti-miR-944 (blue), ΔNp63 siRNA (yellow), and a combination of anti-miR-944 and ΔNp63 siRNA (red) were identified and their expression was compared with that with the scramble treatment. (**C**) The Venn diagrams show the common DEGs in ME-180 cells treated with anti-miR-944 (blue), ΔNp63 siRNA (red), and a combination of anti-miR-944 and ΔNp63 siRNA (green). Significant DEG gene ontology (GO) terms for (**D**) anti-miR-944, (**E**) ΔNp63 siRNA, and (**F**) a combination of anti-miR-944 and ΔNp63 siRNA were plotted, and higher −log *p*-values were found to be more closely related. All GO terms were statistically significant (*p* < 0.05). The top GO-enriched biological processes in each of the three inhibition conditions were classified into 4 clusters using a kappa test and Bonferroni step down analysis (score  =  0.4 and *p* < 0.05). The experiments were repeated in duplicate.

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
