# Peer review of "Synergetic Effects of Intronic Mature miR-944 and ΔNp63 Isoforms on Tumorigenesis in a Cervical Cancer Cell Line"

_ijms, 2020, doi:10.3390/ijms21165612_

Round 1
Reviewer 1 Report
The manuscript ‘Synergetic effects of intronic mature miR-944 and Np63 isoforms on tumorigenesis in cervical cancer’ evaluates the role of miR-944 and Np63 in cell growth and viability through transfection of anti-miR-944 and Np63-siRNA constructs. Overall, the manuscript is well written and provides supportive data that miR-944 and Np63 alter cancerous cells. Minor revision is recommended based on the following comments.
Line 27: ‘Between’ needs to be the same font size and the rest of the document.
Line 147-148: Figure 4 legend description needs to be improved. Figures and their legends should stand alone. Other figures in the manuscript have a more detailed legend of what the figure represents.
Line 149: ‘Anit’ needs to be changed to ‘anti’.
Line 204: ‘Heap’ needs to be changed to ‘heat’.
Figures such as Figure 6, contain images and text that are not clear. For many of the figures the text and images are blurry. Assuming that higher quality images will be uploaded for the final version.
For many of the experiments, it is not clear in the materials and methods how many samples (e.g., cell culture wells) or replicates were evaluated. Please clarify for all experiments.
For this set of experiments, the authors have only evaluated the effects of constructs that inhibit miR-944 and Np63 (anti-miR-944 and Np63-siRNA). Is there previous literature that demonstrates the overexpression of miR-944 and Np63? To properly evaluate the role of the miR and Np63, it would be of interest to know what overexpression of miR-944 and Np63 does.
Author Response
Response to Reviewer 1 Comments
The manuscript ‘Synergetic effects of intronic mature miR-944 and Np63 isoforms on tumorigenesis in cervical cancer’ evaluates the role of miR-944 and Np63 in cell growth and viability through transfection of anti-miR-944 and Np63-siRNA constructs. Overall, the manuscript is well written and provides supportive data that miR-944 and Np63 alter cancerous cells. Minor revision is recommended based on the following comments.
Author’s response: We deeply appreciate for your time and careful review. Based on your comments, we have revised the manuscript accordingly.
Point 1: Line 27: ‘Between’ needs to be the same font size and the rest of the document.
Author’s response: We corrected ‘between’ with the same font size and style.
(Page 1, line 28 in revised manuscript)
Point 2: Line 147-148: Figure 4 legend description needs to be improved. Figures and their legends should stand alone. Other figures in the manuscript have a more detailed legend of what the figure represents.
Author’s response: We corrected and added Figure 4 legend description in detail.
(Page 6, line 160-167 in revised manuscript)
Point 4: Line 149: ‘Anit’ needs to be changed to ‘anti’.
Author’s response: We corrected ‘anit’ to ‘anti’.
(Page 6, line 161 in revised manuscript)
Point 5: Line 204: ‘Heap’ needs to be changed to ‘heat’.
Author’s response: We corrected ‘heap’ to ‘heat’
(Page 8, line 217 in revised manuscript)
Point 6: Figures such as Figure 6, contain images and text that are not clear. For many of the figures the text and images are blurry. Assuming that higher quality images will be uploaded for the final version.
Author’s response: Based on reviewer’ comments, we adjusted to a high quality of all figures including figure 6.
Point 7: For many of the experiments, it is not clear in the materials and methods how many samples (e.g., cell culture wells) or replicates were evaluated. Please clarify for all experiments.
Author’s response: We added the number of replicates in the each figure legend.
(For figure 1, Page 3, line 100 in revised manuscript)
(For figure 2, Page 4, line 118 in revised manuscript)
(For figure 3, Page 5, line 141-142 in revised manuscript)
(For figure 4, Page 6, line 166 in revised manuscript)
(For figure 5, Page 7, line 191 in revised manuscript)
(For figure 6, Page 9, line 259 in revised manuscript)
Point 8: For this set of experiments, the authors have only evaluated the effects of constructs that inhibit miR-944 and Np63 (anti-miR-944 and Np63-siRNA). Is there previous literature that demonstrates the overexpression of miR-944 and Np63? To properly evaluate the role of the miR and Np63, it would be of interest to know what overexpression of miR-944 and Np63 does.
Author’s response: For miR-944, previous studies demonstrates that miR-944 promotes onco-miR activating migration, proliferation, and invasion in several cancer such as endometrial cancer, breast cancer, lung cancer and cervical cancer. We added previous studies for ‘overexpression of miR-944 in clinical setting’ in discussion section.
- “Furthermore, the expression levels of miR-944 in cervical tissues were significantly higher than those in normal tissues and high miR-944 expression was associated with bulky tumor size, advanced stage, lymph node metastasis, and poor survival”
(Page 10, line 273-275 in revised manuscript)
For ΔNp63, ΔNp63 has an oncogenic function and overexpression of ΔNp63 is reported to be related to cell survival, migration. However, several studies have reported that down-regulation of ΔNp63 is also related to poorly differentiated tumors, and to poor prognosis. We also added previous study for ‘overexpression of ΔNp63 in clinical setting’ in discussion section.
- ΔNp63 was also highly expressed in several cancer tissues such as esophageal squamous cell carcinoma, and cervical cancer with poor prognosis
(Page 10, line 285-286 in revised manuscript)
Reviewer 2 Report
The article by Kim et al. shows correlation between miR-944 expression levels and cervical cancer stage, lymph node metastasis and survival. In general manuscript reads well, but requires major revision as for now it contains far too many weaknesses.
Major concerns:
- Results presented in the figures 2 and 3 are unconvincing. Effects on viability of 20% is nothing. Authors should perform additional experiments in prolonged times: 48, 78, 96 h. Reviewer suggests to decrease also the number of cells plated. Additionally, authors should perform also colony forming assay to show how the inhibition of these two molecules influence the ability of the cells to form colonies.
- In relation to the figures 2 and 3, control values do not have SD indicated, and this suggest that authors normalized results of each experiments before they combine all replicates. This is inappropriate and it could distort the results. Authors should work on raw data and after statistical analysis they can normalize them to the controls.
- Authors use inappropriate statistical tests. All results that were obtained using ME-180 cell line should be reanalyzed using ANOVA following post-hoc test (e.g. Tukey or Dunn’s).
- Materials and method section. 4.11. RNA sequence analysis. Authors did not include many important information: how adapter trimming was performed? Prior to DEG analysis, how the gene expression statistics were analyzed? Which r package was used for bioinformatic comparison between samples?
Minor concerns:
- Figure 1. Please include the median values to the figures.
- Figure 4C and Figure 5E. Authors should present uncut original blot scans in supplementary materials.
- Authors should include the number of replicates in the each figure legend.
- Results section. Please include Venn’s diagram to show whether there are any common DEGs between miR-944 and deltaNp63 cohorts.
- Discussion section. Line 250, authors claim: “Firstly, tumorigenesis induced by the inhibition (…).” Literally, tumorigenesis is the formation of a cancer and authors did not investigate this process.
- Materials and methods section, 4.7 subsection. Please include all antibodies used in this study with the catalog numbers included.
- Please include the datasets showing all DEG’s in exel file in supplemenatary materials. The same also applies for the ontological analysis.
Author Response
Response to Reviewer 2 Comments
The article by Kim et al. shows correlation between miR-944 expression levels and cervical cancer stage, lymph node metastasis and survival. In general manuscript reads well, but requires major revision as for now it contains far too many weaknesses.
Author’s response: We deeply appreciate for your time and careful review. Based on your comments, we have revised the manuscript accordingly.
Major concerns:
Point 1: Results presented in the figures 2 and 3 are unconvincing. Effects on viability of 20% is nothing. Authors should perform additional experiments in prolonged times: 48, 78, 96 h. Reviewer suggests to decrease also the number of cells plated. Additionally, authors should perform also colony forming assay to show how the inhibition of these two molecules influence the ability of the cells to form colonies.
Author’s response: We performed additional experiments in prolonged time: 48h and 72h and added the data in figure 2 and 3. We also added trypan blue assay in figure 3.
(For figure 2, Page 3, line 109-112, 115-117 in revised manuscript)
(For figure 3, Page 4-5, line 132-135, 139-141 in revised manuscript)
However, we did not perform colony forming assay. As reviewer’s comments, we mentioned limitation of needs for additional investigation including colony forming assay.
(Page 11, line 327-328 in revised manuscript)
Point 2: In relation to the figures 2 and 3, control values do not have SD indicated, and this suggest that authors normalized results of each experiments before they combine all replicates. This is inappropriate and it could distort the results. Authors should work on raw data and after statistical analysis they can normalize them to the controls.
Author’s response: We thank the reviewer for important comments. For figure 2 and 3, we showed SD indicated in control group.
Point 3: Authors use inappropriate statistical tests. All results that were obtained using ME-180 cell line should be reanalyzed using ANOVA following post-hoc test (e.g. Tukey or Dunn’s).
Author’s response: We reanalyzed using ANOVA with post-hoc test (Tukey’s multiple comparisons test) for differences between four or more independent groups with figure 3, figure 4, figure 5.
Point 4: Materials and method section. 4.11. RNA sequence analysis. Authors did not include many important information: how adapter trimming was performed? Prior to DEG analysis, how the gene expression statistics were analyzed? Which r package was used for bioinformatic comparison between samples?
Author’s response: In this study, Targeted sequencing was performed using AmpliSeq Trascriptome human gene expression kit. We added primary analysis for AmpliSeq sequencing data with reference [43,44]. DEG analysis was performed using transcriptome analysis console (TAC) software (ThermoFisher) designed for QC, normalization, statistical tests for differential expression. We also added site as reference [45] which exist information, manual of TAC software.
(Page 14, line 465-470 in revised manuscript)
Minor concerns:
Point 5: Figure 1. Please include the median values to the figures.
Author’s response: We included the mean values to the figures to show difference between groups.
(Page 2, line 90-92 in revised manuscript)
Point 6: Figure 4C and Figure 5E. Authors should present uncut original blot scans in supplementary materials.
Author’s response: We added uncut original blot scans in supplementary materials (Figure S2).
Point 7: Authors should include the number of replicates in the each figure legend.
Author’s response: We added the number of replicates in the each figure legend.
(For figure 1, Page 3, line 100 in revised manuscript)
(For figure 2, Page 4, line 118 in revised manuscript)
(For figure 3, Page 5, line 141-142 in revised manuscript)
(For figure 4, Page 6, line 166 in revised manuscript)
(For figure 5, Page 7, line 191 in revised manuscript)
(For figure 6, Page 9, line 259 in revised manuscript)
Point 8: Results section. Please include Venn’s diagram to show whether there are any common DEGs between miR-944 and deltaNp63 cohorts.
Author’s response: we added Venn’s diagrams to show whether there are any common DEGs between miR-944 and deltaNp63 cohorts in figure 6C and exel file in supplementary data.
Point 9: Discussion section. Line 250, authors claim: “Firstly, tumorigenesis induced by the inhibition (…).” Literally, tumorigenesis is the formation of a cancer and authors did not investigate this process.
Author’s response: Based on the reviwer’s comments, we revised ‘tumorignesis’ to ‘apoptosis and migration’ to avoid the confusion for further readers.
(Page 10, line 267 in revised manuscript)
Point 10: Materials and methods section, 4.7 subsection. Please include all antibodies used in this study with the catalog numbers included.
Author’s response: We added the catalog numbers for all antibodies used in this study in materials and methods section (4.8 subsection in revised manuscript).
(Page 12, line 391-395 in revised manuscript)
Point 11: Please include the datasets showing all DEG’s in exel file in supplemenatary materials. The same also applies for the ontological analysis.
Author’s response: we added dataset for DEG analysis data and ontological analysis data in supplementary materials.
Reviewer 3 Report
Dear Authors,
Please find below recommendations for additional experiments and minor changes I believe are needed prior to publication:
Minor:
- The authors used one specific cell line to investigate the synergistic effects of miR-944 and ΔΝp63 inhibition. The title should be changed to reflect this. " Synergetic effects of intronic mature miR-944 and ΔNp63 isoforms on tumorigenesis in a cervical cancer cell line "
- In Figure 3A, the result is not quantifiable. Did the authors perform viable cell count in the microscopy images in some manner? In addition to the WST assay, Trypan Blue would also be useful.
- Figure 4A: The authors should quantify early apoptosis as well (bottom right quadrant).
- Figure 4C: The authors should quantify the Western Bolt bands with densitometry to show if the combination effect is greater than the single agents alone.
- Figure 5E: Densitometry is also needed in this Western Blot.
Major:
- The authors did not show the levels of miR-944 or ΔNp63 in the cell line they are investigating. In my opinion,this experiment should be performed before and after inhibition with siRNA.
- Figure 2A and 2B: The decrease in cell viability is not very robust, reaching only 80% at 100 nM of treatment with either inhibitor. Why did the authors not increase the incubation time? I believe a time-depended experiment perhaps 48 and 72 hours is also needed.
- Figure 3B: Also the decrease in viability here is not robust. Why not increase the time to 48 or 72 hours?
Author Response
Response to Reviewer 3 Comments
Dear Authors,
Please find below recommendations for additional experiments and minor changes I believe are needed prior to publication:
Author’s response: We deeply appreciate for your time and careful review. Based on your comments, we have revised the manuscript accordingly.
Major:
Point 1: The authors did not show the levels of miR-944 or ΔNp63 in the cell line they are investigating. In my opinion,this experiment should be performed before and after inhibition with siRNA.
Author’s response: We added the data for the levels of miR-944 or ΔNp63 in the cell line before and after inhibition with anti-miR-944 or ΔNp63 siRNA in supplementary figure 1. we also added the description in result section.
(Page 3, line 104-108 in revised manuscript)
Point 2: Figure 2A and 2B: The decrease in cell viability is not very robust, reaching only 80% at 100 nM of treatment with either inhibitor. Why did the authors not increase the incubation time? I believe a time-depended experiment perhaps 48 and 72 hours is also needed.
Author’s response: We performed additional experiments in prolonged time: 48, 72 and added the data in figure 2.
(Page 3, line 109-112, 115-118 in revised manuscript)
Point 3: Figure 3B: Also the decrease in viability here is not robust. Why not increase the time to 48 or 72 hours?
Author’s response: We performed additional experiments in prolonged time: 48, 72 h and added the data in figure 3.
(Page 4-5, line 132-135, 139-140 in revised manuscript)
Minor:
Point 1: The authors used one specific cell line to investigate the synergistic effects of miR-944 and ΔΝp63 inhibition. The title should be changed to reflect this. " Synergetic effects of intronic mature miR-944 and ΔNp63 isoforms on tumorigenesis in a cervical cancer cell line "
Author’s response: We changed the title as “Synergetic effects of intronic mature miR-944 and ΔNp63 isoforms on tumorigenesis in a cervical cancer cell line”
Point 2: In Figure 3A, the result is not quantifiable. Did the authors perform viable cell count in the microscopy images in some manner? In addition to the WST assay, Trypan Blue would also be useful.
Author’s response: We added viable cell count data (trypan blue test) in figure 3.
(Page 4-5, line 132-135, line 355-360 in revised manuscript)
Point 3: Figure 4A: The authors should quantify early apoptosis as well (bottom right quadrant).
Author’s response: We added early apoptosis data (Annexin V+/PI-) in figure 4B.
Point 4: Figure 4C: The authors should quantify the Western Bolt bands with densitometry to show if the combination effect is greater than the single agents alone.
Author’s response: We quantified the western blot band Figure 4C with densitometry to show whether the combination effect is greater than single agents.
Point 5: Figure 5E: Densitometry is also needed in this Western Blot.
Author’s response: We also quantified the western blot band for Figure 5E with densitometry to show whether the combination effect is greater than single agents.
Round 2
Reviewer 2 Report
The authors reacted in a good way to my criticism. Thus, I have no further comments.
Author Response
The authors reacted in a good way to my criticism. Thus, I have no further comments.
Author’s response: We would like to appreciate for your positive decision for our manuscript.
Reviewer 3 Report
Dear authors,
Please find below minor suggestions:
Minor:
- Please correct in title: …in a cervical cancer cell line
- The basal expression levels of miR-944, ΔNp63 and TAp63 before transfection as shown in Figure S1 should be compared with at least one more cervical cell line or with normal cells. Does the correlation exist between miR-944 levels and ΔNp63 levels in the ME-180 cell line compared to other cell lines like it does in cervical cancer tissues as shown in Figure 1? This may be addressed with evidence from the literature.
- Also please correct the title of the x axis of Figure S1 panel C, to indicate TAp63 as per the text.
Author Response
Dear authors,
Please find below minor suggestions:
Minor:
Author’s response: We would like to appreciate for important comments.
- Please correct in title: …in a cervical cancer cell line
Author’s response: We corrected the title based on reviewer’s comment.
‘Synergetic effects of intronic mature miR-944 and ΔNp63 isoforms on tumorigenesis in a cervical cancer cell line’
- The basal expression levels of miR-944, ΔNp63 and TAp63 before transfection as shown in Figure S1 should be compared with at least one more cervical cell line or with normal cells. Does the correlation exist between miR-944 levels and ΔNp63 levels in the ME-180 cell line compared to other cell lines like it does in cervical cancer tissues as shown in Figure 1? This may be addressed with evidence from the literature.
Author’s response: For basal expression levels of ΔNp63 and TAp63, we previously investigated five cell lines including ME-180 with references [22]. For basal expression levels of miR-944, we added description with reference [25].
(Page 10, line 267-270 in revised manuscript)
We also added the needs for further investigation with various cancer cell lines for correlation between miR-944 and ΔNp63.
(Page 11, line 332 in revised manuscript)
- Also please correct the title of the x axis of Figure S1 panel C, to indicate TAp63 as per the text.
Author’s response: We showed the title of the Y axis of Figure S1C to indicate TAp63 (TAp63 expression levels). For the title of the x axis of Figure S1C, we showed groups for negative control and treatment with ΔNp63 siRNA. We confirmed specific inhibition effect of ΔNp63 siRNA.